# Effects of glycosaminoglycan content in extracellular matrix of donor cartilage on the functional properties of osteochondral allografts evaluated by micro-CT non-destructive analysis

**Yong Jun Jin**[1,2], **Do Young Park**[1,2], **Sujin Noh**[3], **HyeonJae Kwon**[2,4], **Dong Il Shin**[2,4], **Jin Ho Park**[2,4], **Byoung-Hyun Min**[1,2,4]*

1 Department of Orthopedic Surgery, School of Medicine, Ajou University, Suwon, Republic of Korea, 2 Cell Therapy Center, Ajou University Medical Center, Suwon, Republic of Korea, 3 Department of Biomedical Sciences, Graduate School of Ajou University, Suwon, Republic of Korea, 4 Department of Molecular Science and Technology, Ajou University, Suwon, Republic of Korea

* bhmin@ajou.ac.kr

## Abstract

Osteochondral allograft (OCA) is an important surgical procedure used to repair extensive articular cartilage damage. It is known that chondrocyte viability is crucial for maintaining the biochemical and biomechanical properties of OCA, which is directly related to the clinical success of the operation and is the only standard for preoperative evaluation of OCA. However, there is a lack of systematic research on the effect of the content of cellular matrix in OCA cartilage tissue on the efficacy of transplantation. Therefore, we evaluated the effect of different GAG contents on the success of OCA transplantation in a rabbit animal model. Each rabbit OCA was treated with chondroitinase to regulate glycosaminoglycan (GAG) content in the tissue. Due to the different action times of chondroitinase, they were divided into 4 experimental groups (including control group, 2h, 4h, and 8h groups). The treated OCAs of each group were used for transplantation. In this study, transplant surgery effects were assessed using micro-computed tomography (μCT) and histological analysis. Our results showed that tissue integration at the graft site was poorer in the 4h and 8h groups compared to the control group at 4 and 12 weeks in vivo, as were the compressive modulus, GAG content, and cell density reduced. In conclusion, we evaluated the biochemical composition of OCAs before and after surgery using μCT analysis and demonstrated that the GAG content of the graft decreased, it also decreased during implantation; this resulted in decreased chondrocyte viability after transplantation and ultimately affected the functional success of OCAs.

## 1. Introduction

Articular cartilage defect is a common orthopedic disease [1, 2]. Cartilage is limited in its ability to repair itself due to insufficient vascular supply and the inability of differentiated cell

**Data Availability Statement:** All relevant data are within the paper and its Supporting information files.

**Funding:** This study was supported by the Korea Health Technology R&D Project (HI17C2191) through the Korea Health Industry Development Institute, funded by the Ministry of Health & Welfare. We have no potential conflict of interest relevant to this article. The funders had no role in study design, data collection and analysis, decision to publish, or preparation of the manuscript. This study was approved by the IACUC (IACUC No.2021-0030) at the Laboratory of Animal Research at Ajou University Medical Center.

**Competing interests:** The authors have declared that no competing interests exist.

populations to respond to injury [3, 4]. Therefore, the treatment of articular cartilage defects remains a challenging clinical problem. Surgical options currently used clinically for articular cartilage repair include microfracture, autologous chondrocyte implantation, osteochondral autograft, and allograft transplantation [5]. Especially for large areas of full-thickness articular cartilage lesions ($>3cm^2$), osteochondral allograft (OCA) transplantation has unique advantages [6–8]. OCA contains available chondrocytes and hyaline cartilage-related matrix for direct transplantation into the defect site [9]. In the most recent data for fresh OCA transplantation on the femoral condyle, the 5-year survival rate of chondrocytes was between 90% and 95%, the 10-year survival rate was around 85%, and the 15-year survival rate was more than 70% [10, 11]. Furthermore, the survival rate of chondrocytes is an important factor in the long-term functional maintenance of OCA [12, 13].

Although many studies have demonstrated clinical efficacy of OCA transplantation in osteochondral defects, procedures such as pre-transplant donor matching and the diagnosis of pathogen that could transmit required at least 2 weeks, which limits the transplantation efficiency [14, 15]. Studies have revealed that chondrocyte viability is critical for maintaining the biochemical and biomechanical characteristics of OCA, which the only criterion for preoperative assessment of OCA and directly related to the clinical success of the surgery [13, 16–19]. Therefore, a number of graft storage methods including cryopreservation have been suggested to ensure the functional integrity of chondrocytes and ECM in OCA [16, 20–23].

Articular cartilage ECM regulates chondrocyte function through cell-matrix interactions, building cytoskeleton and integrin-mediated signal transduction [24]. Given its chondroinductive properties, many decellularized cartilage matrices are used for cartilage regeneration, such as Zimmer's Chondrofix® osteochondral allograft products [25]. Therefore, the ECM content of donor cartilage is also an important factor affecting the success of implantation in OCA transplantation. Under normal physiological conditions of articular cartilage, homeostasis and turnover of the ECM is dependent on the responses of chondrocytes to self and paracrine anabolic and catabolic pathways [26, 27]. For example, chondrocytes regulate the synthesis of proteoglycans and type II collagen by controlling growth factors and cytokines [28, 29]. Likewise, the ECM is a complex network that surrounds chondrocytes and is highly fluid. It is composed of protein and proteoglycan components and generates biochemical and biomechanical signals to chondrocytes, thereby regulating cellular processes including growth, differentiation, migration, homeostasis, survival and morphogenesis [30]. An integrated ECM structure maintains the stability of the extracellular microenvironment, thereby preventing cellular damage and apoptosis [31]. Therefore, the integrity of the ECM of the cartilage tissue in the graft is also an important factor for successful OCA function after transplantation.

GAG is an important component of articular cartilage ECM, and it is an important detection marker in various physiological and pathological stages of articular cartilage [32, 33]. Among the biomechanical functions of the ECM, GAGs form a stress-strain fluid flow microenvironment by regulating osmotic pressure and chemical expansion stress effects in the tissue, and play a buffering mechanism in cartilage tissue, ultimately preventing cell damage [34–36]. Moreover, the negatively charged GAG binds to interstitial water to provide bio-compression stiffness [37, 38]. This function maintains the stability of the extracellular microenvironment when the cartilage tissue is subjected to continuous pressure load [39, 40]. Interestingly, this negatively charged characteristic of GAG provides us with a non-destructive analysis method using μCT to calculate the GAG content in the graft, thus confirming the results of several previous studies [38, 41–43]. This leads to novel directions for a new detection mode for the process of graft quality control.

In general, achieving successful results after OCA implantation is closely related to donor cartilage cells, but the role played by GAG content in the donor cartilage tissue has yet to be

fully elucidated. We hypothesized that maintaining a high GAG content in the graft might be an important factor for the functional success of OCA after implantation. Therefore, the purpose of this study was to determine the importance of the GAG content of the donor cartilage tissue for implantation success in rabbit model by using the μCT non-destructive analysis method.

## 2. Materials and methods

### 2.1 Experimental design

Fig 1 shows an overview of the experimental strategy. OCAs with a diameter of 3.5 mm were obtained from femoral trochlear parts of bilateral knee in New Zealand white rabbits (weight 2.5–3.0 kg), and the grafts were then treated with chondroitinase ABC (Sigma-Aldrich, USA). The experiments were divided into four groups according to the time of chondroitinase treatment; Group 1: control (grafts were not treated with chondroitinase; untreated group), Group 2: 2h group (grafts received 2 hours of chondroitinase treatment), Group 3: 4h group (grafts received 4 hours chondroitinase treatment), and Group 4: 8h group (grafts received 8 hours chondroitinase treatment). This experiment is divided into two separate parts. In the previous part, the correlation between μCT values and GAG content were determined by analyzing μCT value, GAG content, cell viability and histological staining of OCA at in vitro study. As a preliminary in vivo study (with the following part of the same surgical approach), the OCA of the graft site was obtained and analyzed for μCT and GAG content and at 4 and 12 weeks postoperatively. In the latter part, μCT analysis was performed on OCA obtained from donors and then implanted into subjects to demonstrate the influence of GAG content on the functional success of the graft. Biomechanics, radiology, histology, and cell viability were examined at 4 and 12 weeks postoperatively to determine the effect of OCA transplantation.

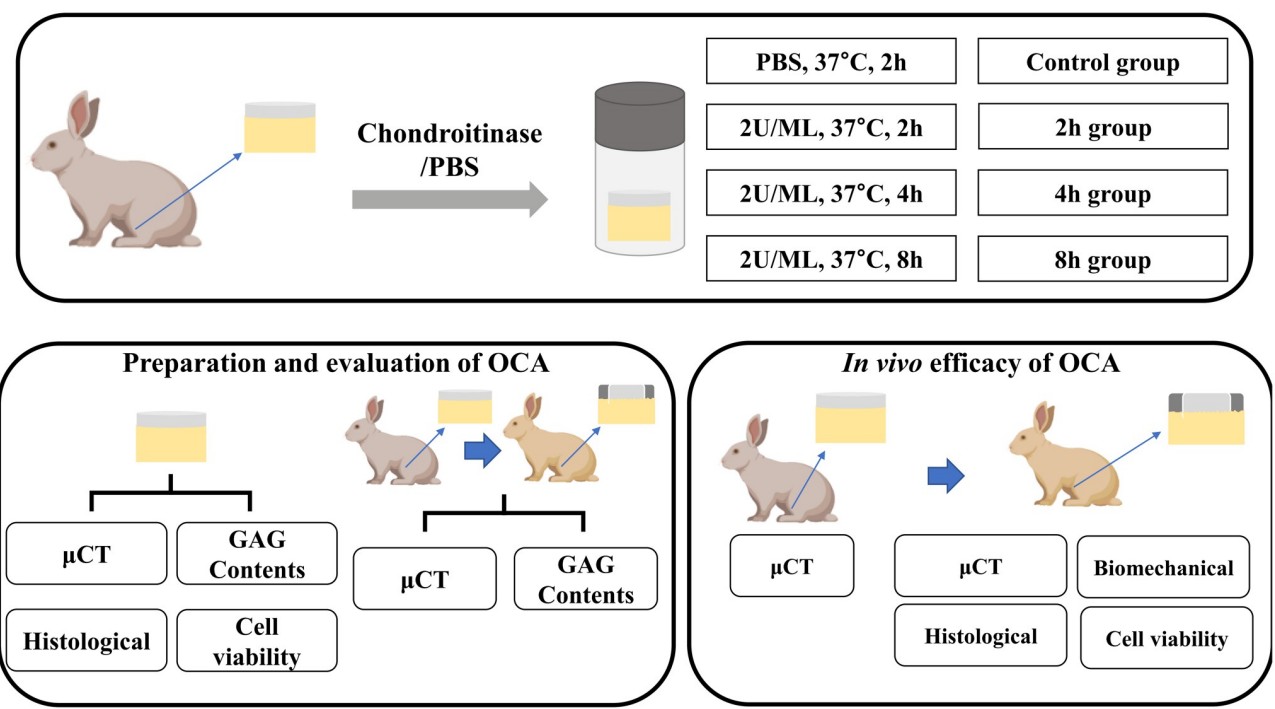

**Fig 1. Simplified diagram of the overall research design.**

## 2.2 Preparation and characterization of OCA

All experimental protocols involving animals were conducted under IACUC approval (IACUC No. 2021–0030) at the Laboratory of Animal Research at XXX University Medical Center. A total of 68 male New Zealand white rabbits (KOATECH, Pyeongtaek, Korea) were used; 44 were used for in vitro research, while the remaining 24 were used for preliminary in vivo tests. The rabbits were 12 weeks old and weighed 2.5–3.0 kg. OCAs with a diameter of 4 mm were obtained from the femoral trochlear parts of the bilateral knee joints of adult New Zealand male rabbits. After isolation, they were stored in phosphate-buffered saline (PBS) solution at room temperature. Each graft was then soaked to a medium containing 1 unit/ml chondroitinase at 37˚C for 2, 4, or 8 hours according to the assigned group (while the control group received an 8-hour treatment of PBS solution). After the treatment, each OCA was washed for 5 minutes each in PBS solution three times. For preliminary in vivo experiments, OCAs were obtained from 12 donor rabbits and treated with chondroitinase as described above, then each group of OCAs were implanted into 12 subject rabbits. To analyze the OCA samples in terms of μCT and GAG content, the graft sites were obtained 4 and 12 weeks after surgery.

**2.2.1 Radiographic analysis of OCA.**   Iobrix Injection contrast agent (TAEJOON, South Korea) was applied to the OCA for 3 hours at room temperature. Then, the μCT value of OCA was determined with a microcomputer tomography device (CT, Skyscan 1076, Skyscan, Belgium). The CT parameters were set to a resolution of 18.22 m pixels with an exposure time of 300 ms using a 40 kV energy source and a 200 mA current. The imaging and data analysis were conducted using Skyscan software (Skyscan, Belgium). This study used Hounsfield units (HU), which range from 0 to 3000 [38], to measure the average CT attenuation. To remove the components of the contrast agent in the tissue, each OCA was washed three times in PBS solution for five minutes each time following the μCT analysis. Subsequently, we analyzed OCA for GAG content, cell viability, and histological staining.

**2.2.2 Measurement of GAG contents in OCA.**   The cartilage tissue that separated from each OCA (n = 4) was freeze-dried and incubated with a papain digestion solution (5 mM L-cysteine, 100 mM Na2HPO4, 5 mM EDTA, and 125 mg/ml papain type III: Sigma-Aldrich, Missouri, USA) overnight at 60˚C. At this point, a 1,9-dimethyl- methylene blue (DMMB) Blyscan™ Glycosaminoglycan test kit was used to determine the total sGAG content (b1000, bicolor, UK) followed by manufacturer's instruction.

**2.2.3 Comparison of μCT value and GAG quantification.**   In the present study, we reproduced the measurement of the GAG content of OCA by μCT analysis. In order to confirm the correlation between the μCT value and the GAG content after implantation of OCA in vivo, the following experiments were performed. Assuming that the mean CT value reflects the GAG content of the tissue, μCT analysis and GAG content detection were performed on each group of grafts (n = 64, including 40 samples from in vitro experiments and 24 samples from 4 and 12 weeks of preliminary in vivo experiments). The experimental method described above was used to measure the μCT value and GAG content of each graft; the relationship between the two values was analyzed, and $R^2$ (determination coefficient) was calculated to determine the applicable range and degree of effectiveness.

**2.2.4 Cell viability measurement of OCA graft.**   After μCT imaging, LIVE/DEAD cell staining (L3224, Thermo Fisher, USA) was conducted to determine the viability of the chondrocytes in each set of grafts (n = 4). A scalpel (1 mm) was used to cut a full-thickness cross-sectional slice of the cartilage. The cartilage slice was treated with the LIVE/DEAD working solution (2 μM calcein-AM and 4 μM ethidium homodimer-1 in PBS), which was then incubated for 1 hour at room temperature. Each cartilage slice was washed three times in PBS

before imaging by confocal microscopy (Zeiss LSM510 Meta). The percentage of chondrocyte viability for each graft was quantified using digital image analysis. As the mean viable chondrocyte density (cells/mm2) assessed from this graft is reported as the number of viable chondrocytes per graft [13], the area of the cartilage tissue was measured and the ratio of the number of live chondrocytes to the area of the cartilage was used to calculate the viable chondrocyte density.

**2.2.5 Histologic analysis of OCA.** After μCT imaging, specimens were preserved for 7 days at room temperature in Neutral Buffered Formalin (NBF) at a concentration of 10%. Before being embedded in paraffin wax, the samples were decalcified with 5% nitric acid for 3 days (n = 4). 4 μm thick slides of OCA were prepared and stained with Safranin O/Fast green. To determine the extent of cartilage repair, sections from five samples per group were scored blindly by three independent observers using the Histological Scoring System for Assessment of Cartilage Repair [44].

**2.2.6 Biomechanical analysis.** To evaluate the biomechanical characteristics of OCA in each group, samples were collected (n = 4), and compressive tests were performed in a versatile biomechanical system machine (testXpert III testing software, Zwick Roell, ProLine, Germany) equipped with a 1000N load cell. A fully automated series of compressive-relaxation steps (step 5μm, velocity 1 μm/min) was repeated up to 50% strain. After a phase test, 60 seconds of relaxation was set as the criterion for the beginning of a new step. For each sample, the test was repeated three times, and the stress-strain curve were made. Finally, the compressive modulus was obtained from the slope of the 5–15% linear part of the stress-strain curve [45].

## 2.3 Preoperative evaluation

OCAs with dimensions of 3.5 mm diameter x 2.5 mm height were obtained from the donor's rabbit knee femoral trochlear parts using a special biopuncher. After conducting an evaluation by μCT analysis and after three washes with PBS solution, OCAs were treated with chondroitinase and evaluated by μCT analysis; finally, after 3 washes with PBS solution, the OCA implantation procedure was prepared. The above steps were all carried out under sterile conditions.

**2.3.1 Animal model and surgical procedure.** A total of 100 male New Zealand white rabbits (KOATECH, Pyeongtaek, Korea) aged 12 weeks (body weight 2.5–3.0 kg) were used. 50 rabbits were donors and the remaining 50 were operated on bilateral knee joints. 50 rabbits were divided into four experimental groups: Control group, 2h group, 4h group, and 8h group. The biomechanical tests (n = 5), histological analysis (n = 5) was performed in each group at 4 and 12 weeks after surgery, and cell survival analysis was performed 12 weeks after surgery (n = 5). The surgical procedure can be briefly described as follows: Each rabbit was anesthetized using Zoletil (10 mg/kg of zolazepam and tiletamine, Virbac Laboratoire) and Rompun (10 mg/kg of xylazine hydrochloride, Bayer Korea) (1:2 ratio, 1 mL/kg). After a skin incision under sterile conditions, a medial parapatellar approach was taken. A cylindrical defect with a diameter of 3.5 mm and a depth of 2.5 mm was produced in the exposed femoral trochelear, and the allograft system calibration press-fit method was used to randomly implant each group of prepared OCAs [46]. Each rabbit received an intramuscular injection of antibiotic (cefotaxime 1g in a dose of 150mg/kg/day) and analgesic (ketorolac 60 mg/day) for three days postoperatively. At 4 and 12 weeks after surgery, the rabbits were sacrificed, and knee samples were collected for further evaluation.

**2.3.2 Gross morphological analysis.** Gross morphological evaluation of each sample was performed immediately after animal sacrifice to evaluate defect filling, surface smoothness, and tissue integration. Samples were blindly scored by three independent evaluators based on the International Cartilage Research Society (ICRS) scoring system.

**2.3.3 μCT and μCT GAG quantification analyses.** At 4 and 12 weeks after surgery, all samples were collected and processed with Iobrix Injection contrast agent (TAEJOON, Korea), then used for μCT imaging and analysis of data by Skyscan software (Skyscan, Belgium) to measure HU at the transplant site value. Further, the relative GAG content of each group was calculated by the linear regression diagram of μCT value and GAG content. The contrast agent was then washed with PBS solution to prepare for the subsequent analysis experiments.

## 2.4 Statistical analysis

In this study, all quantitative datasets were expressed as mean ± standard deviation (SD). Shapiro-Wilk test was used to confirm normality of all samples. Analysis of variance (ANOVA) was performed on standardized samples to compare parametric data between groups. Kruskal-Wallis test with Dunn's test was used to compare multiple groups when the data are not normally distributed. Two-way ANOVA and Bonferroni test were applied in the presence of two independent variables. All data analyses were performed using GraphPad Prism 8 (GraphPad Software, La Jolla, CA, USA) to determine whether the results for the various datasets were statistically significantly different, with a value of $P < 0.05$ considered to represent a significant difference ($^*p < 0.05$, $^{**}p < 0.01$, $^{***}p < 0.001$).

## 3. Results

### 3.1 GAG contents and cell viability on OCAs

Fig 2 showed changes of GAG contents and cell viability after treating chondroitinase at time-dependent manner. After treating chondroitinase for 4 and 8 hours at 37°C, the GAG content was significantly reduced. (Control group—116.01±6.3μg/mg; 2h group—86.85±7.8μg/mg; 4h group—75.15±8.3μg/mg; and 8h group—59.34±5.1μg/mg.) The percentages of GAG contents that normalized to the value of control group were as followed: 2h group—75.1%±8.2%; 4h group—64.9%±7.6%; 8h group—51.3%±5.8% (Fig 2A and 2B, $^*p < 0.05$, $^{***}p < 0.001$). On the contrary, the cell viability was maintained over time. Digital image analysis showed that there were no significant differences in chondrocyte viability or viable chondrocyte density between groups (Fig 2D and 2E).

### 3.2 Comparison of μCT value and GAG content in OCAs

The anionic, Iobrix contrast agent is diffused into the ECM of articular cartilage in inverse proportion to the GAG content. Further, the μCT and histological staining analyses were performed in the same OCA. Fig 3A shows a resulting image of μCT and histological analysis. As the chondroitinase application time increased, μCT value (HU value) was also increased. The increase is most significant in the superficial zone of the articular cartilage tissue, whereas less noticeable in the deep zone. In the Safranin-O-stained histological sections, it can be seen that there was increased GAG degradation in the superficial zone of the tissue, and that the red staining progressively decreased. This is consistent with the results of μCT images (Fig 3A). 64 OCAs are used to compare the μCT values and GAG contents, including in vitro experimental samples and preliminary in vivo experimental samples. With increasing treatment time of chondroitinase, the GAG content of OCA decreases whereas the μCT value increases (Fig 3B and 3C). The graph showed a linearly negative correlation between the CT value and the GAG content, with an $R^2$ value of 0.9062 (Fig 3D). The biomechanical characteristics were evaluated by detecting the compressive modulus of OCA in each group. The results showed that there were no statistically significant differences between the control group and the 2h group. By

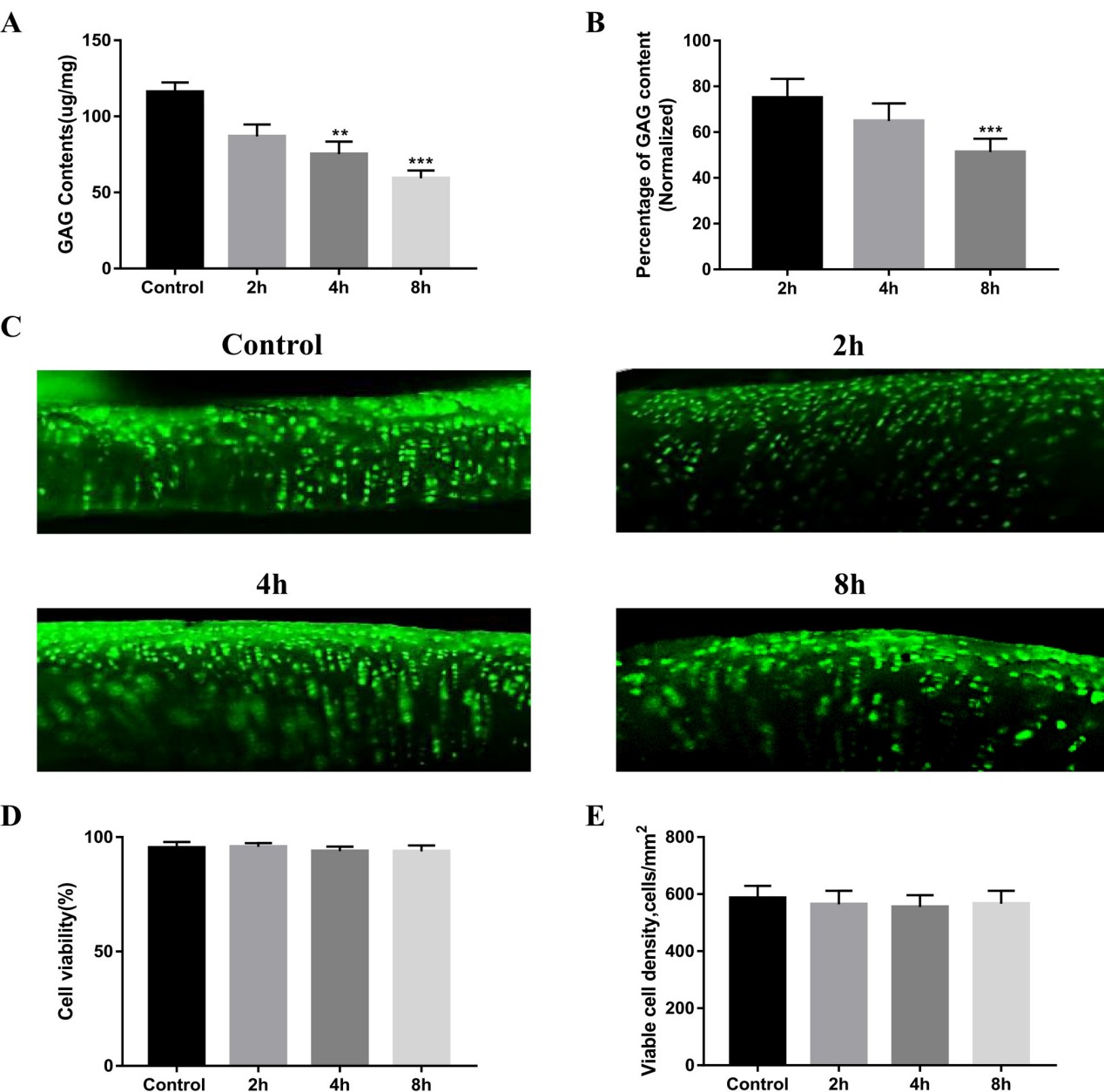

**Fig 2. Changes in GAG contents and cell viability after chondroitinase treatment.** A Biochemical analysis of each group of OCA GAG content and B normalized percentage (n = 10). C Live-dead Confocal microscope images of different groups of OCAs, where dead cells are stained in red and live cells are stained in green. D Quantitative analysis of cell viability and e) viable cell density of each group by Image J software (n = 4). Statistical analysis was conducted using One-way ANOVA test (*p < 0.05, **p < 0.01, ***p < 0.001). Magnification x50.

contrast, the compressive modulus of both the 4h group and the 8h group were significantly lower than those of the control group (Fig 3E, *p < 0.05, ***p < 0.001). In the histological scoring results, the scores of the 4h group and the 8h group were statistically lower than those of the control group. There was no statistically significant difference between the control group and the 2h group (Fig 3F, **p < 0.01).

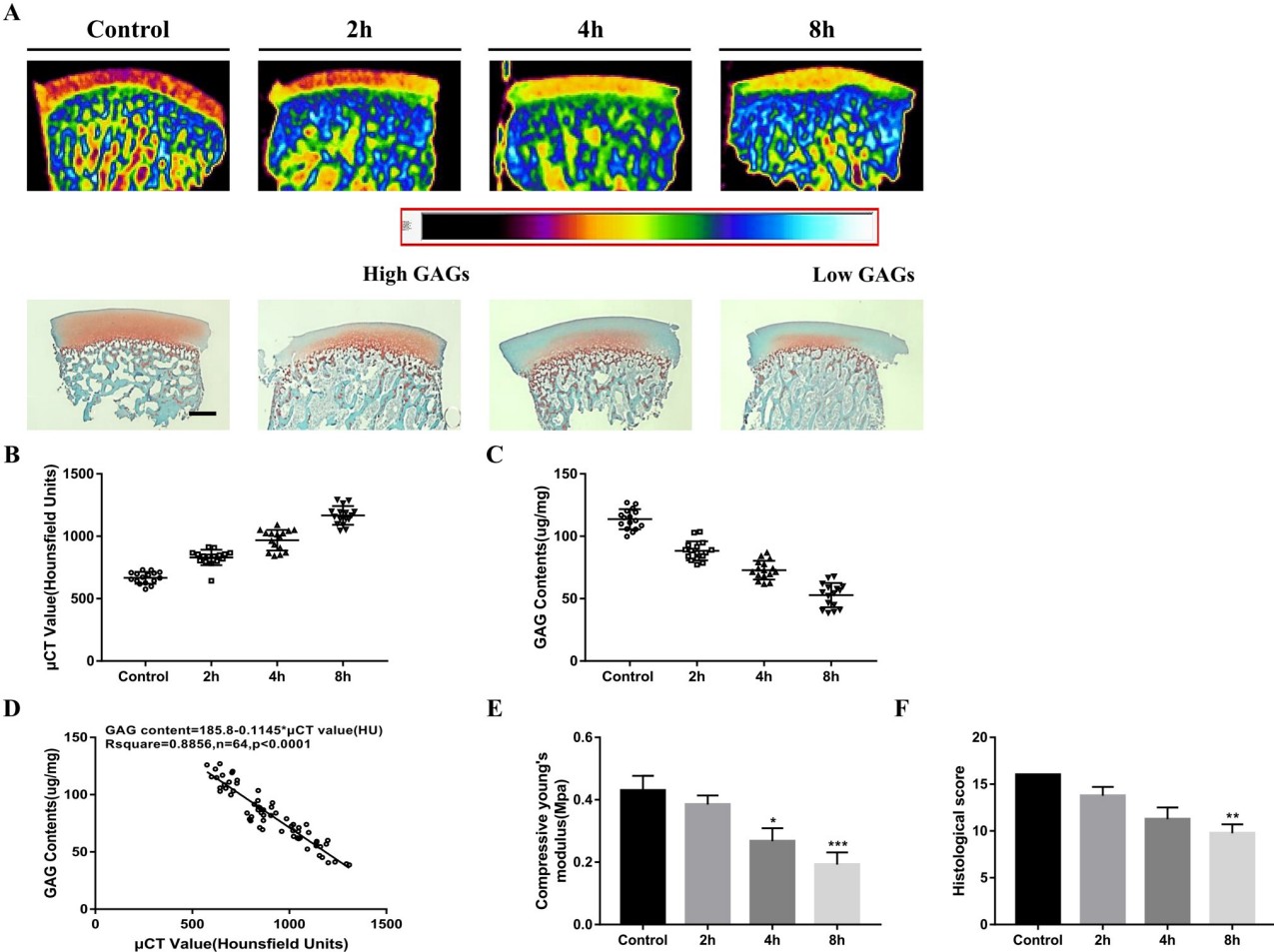

**Fig 3. Changes in GAG content analysis according to μCT analysis in OCA grafts after chondroitinase treatment.** A Representative μCT images and safranin-o-stained images of control and degraded OCA (2h, 4h, and 8h exposure to chondroitinase), the trends of the two analyzed images are inversely proportional to each other (n = 4). Scale bar = 500 μm. B, C Comparison of μCT value and GAG content of each group of OCAs (including in vitro and preliminary in vivo experimental OCAs; 64 in total). With increasing chondroitinase treatment time, the CT value of OCA increased whereas the GAG content decreased. D Linear regression plots: μCT value vs GAG content measured by biochemical assay (n = 64). The $R^2$ in the linear regression graph is 0.8856. E Biomechanical analysis of each group after chondroitinase treatment (n = 4) and F histological score of OCA in each group (n = 4). Statistical analysis was conducted using One-way ANOVA test (*p < 0.05, **p < 0.01, ***p < 0.001).

## 3.3 Preoperative evaluation of OCAs

The preoperative quality evaluation showed that the average μCT value for pretreatment cartilage in each group was as followed: control group—583.9±23.3 HU, 2h group—575.4±48.5 HU, 4h group—563.5±31.0 HU, 8h group—601.9±32.7 HU. Further, the GAG content was calculated based on the relationship between μCT and GAG content: control group—118.9 ±2.7, 2h group—119.9±5.6, 4h group—121.3±3.6, 8h group—116.9±3.7 ug/mg. After chondroitinase treatment, the average μCT value of each group was 846.8±18.7 HU at 2h group, 945.7±30.4 HU at 4h group, and 1089.9±22.7 HU at 8h group. The calculated GAG contents from the estimated formula were 88.8±2.1 at 2h group, 77.5±3.5 at 4h group, and 61.0±2.6 ug/ mg at 8h group. The percentages of GAG content before and after treatment with chondroitinase were: 2h group—74.2±3.6%, 4h group—63.9±2.5%, 8h group—52.2±2.5% (Table 1).

**Table 1. Quality assessment of OCA before animal experiment.**

| Group | HU (Pre-treatment) | μCT GAG quantification | HU (Post-treatment) | μCT GAG quantification | Percentage of GAG contents (%) |
|---|---|---|---|---|---|
| Control | 583.9±23.3 | 118.9±2.7 | - | - | - |
| 2h | 575.4±48.5 | 119.9±5.6 | 846.8±18.7 | 88.8±2.1 | 74.2±3.6% |
| 4h | 563.5±31.0 | 121.3±3.6 | 945.7±30.4 | 77.5±3.5 | 63.9±2.5% |
| 8h | 601.9±32.7 | 116.9±3.7 | 1089.9±22.7 | 61.0±2.6 | 52.2±2.5% |

These results are consistent with the GAG content potential measured by DMMB analysis in Fig 2.

### 3.4 Gross morphological evaluation of OCAs

After 4 weeks implantation, the gross morphological observation results showed that the transplantation sites in the Control group and the 2h group were filled with relatively smooth surfaces and that there was a slight gap between the graft and the host tissue. By contrast, the transplantation sites in the 4h group and 8h group were less filled than the former groups. Moreover, the surface was relatively irregular and the integration with the surrounding host cartilage was inferior to control group. After 12 weeks implantation, the transplantation sites of the Control group and the 2h group were completely repaired by the graft, appearing close to normal tissues. Also, they had a higher degree of fusion with the host tissues. However, tissue defects were still observed in the transplantation sites of the 4h group and the 8h group at 12 weeks, with irregular surfaces and tissue gaps between the host and the graft (Fig 4A). The ICRS assessment results showed that the scores of the control group and the 2h group were higher than those of the 8h group (Fig 4B, **, ## $p < 0.01$) at 4 weeks. There was no statistically significant difference with the scores of the control group and 4h group. At 12 weeks, the scores of the control group and the 2h group were higher than those of the 4h and 8h groups, with a statistical difference (***, ### $p < 0.001$). And the score of the 4h group was also higher than that of the 8h group, which was statistically significant (+++$p < 0.001$).

### 3.5 Biomechanical analysis of OCAs

Biomechanical tests were performed at 4 and 12 weeks postoperatively to demonstrate the mechanical properties of the graft site. At 4 weeks, the compressive moduli of the control group and the 2h group were higher than those of the 4h and 8h groups, with a statistical difference (Fig 4C, *, # $p < 0.05$, **, ## $p < 0.01$). At 12 weeks, it was found that the compressive modulus of the control group and the 2h group was significantly higher than that of the 4h and 8h groups (***, ### $p < 0.001$), and the value of the 4h group was also higher than 8h group, with a statistical difference (+ $p < 0.05$).

### 3.6 Histological and μCT evaluation of OCAs

At 4 weeks, Safranin O staining was performed to reveal cartilaginous ECM deposition in the OCA. The OCAs of the Control group and the 2h group were strongly stained by Safranin O, after which they were close to the color intensity of the surrounding normal tissues, while the staining intensities of the 4h and 8h groups were significantly reduced and very different from those of the surrounding normal tissues. At 12 weeks, the OCAs in the control and 2h groups, which seemed similar to normal cartilage, were found to completely integrate with the surrounding normal tissues. Compared to the control group, the 4h group and the 8h group had less ECM staining, irregular surface, and poor fusion with the surrounding tissue. In the 8h

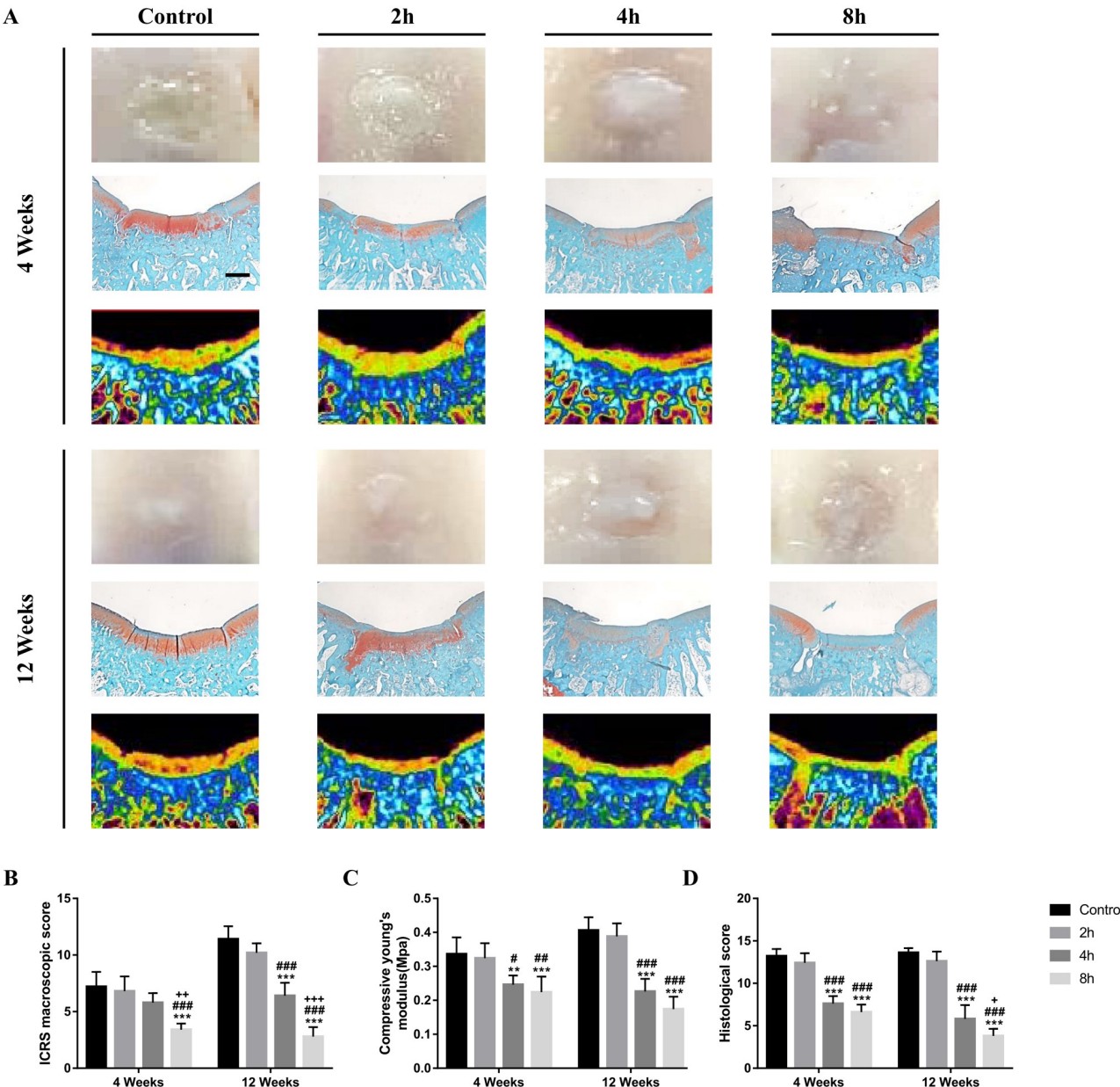

**Fig 4. In vivo experimental analysis of OCA transplant sites. A** Representative gross morphological image, representative Safranin-O staining (n = 5), and μCT analysis images of the OCA transplantation site at 4 and 12 weeks postoperatively. Scale bar = 1 mm. **B** The scores of the International Cartilage Research Society (ICRS) macroscopic evaluation of each transplantation site. Statistical analysis was determined by Kruskal-Wallis test with Dunn's tests. **C** Intergroup comparison of compressive modulus was used to evaluate the biomechanical properties (n = 5). **D** Histological scores for the OCA transplantation site at 4 and 12 weeks after surgery (n = 5). Data represent the means ± SD, *P < 0.05, **P < 0.01, ***P < 0.001 between control group and remaining group. #P < 0.05, ## P < 0.01, ### P < 0.001 between 2h group and remaining group. + P < 0.05, +++ P < 0.001 between 4h group and 8h group. Data were analyzed by two-way ANOVA with Tukey's test if not described.

group, the ECM staining at the transplantation site was severely reduced, the fusion with the surrounding tissue was poor, and tissue collapse was visible at the transplantation site. The histological scores of the control group and the 2h group were significantly higher than those of the other groups at 4 weeks and 12 weeks (Fig 4D, ***, ### p<0.001). And at 12 weeks, the 4h score was higher than the 8h score, and there was a statistical difference (+p<0.05).

**Table 2. Quality assessment of OCA after animal experiment.**

| Time | Group | HU (Post-surgery) | μCT GAG quantification | Percentage of GAG contents (%) |
|---|---|---|---|---|
| 4 weeks | Control | 628.0±30.1 | 113.9±3.4 | 95.8±3.7% |
| | 2h | 854.2±23.4 | 88.0±2.7 | 73.4±2.5% |
| | 4h | 1052.6±45.3 | 65.3±5.2 | 53.9±5.5% |
| | 8h | 1158.3±43.2 | 53.2±4.9 | 45.5±4.5% |
| 12weeks | Control | 621.2±35.8 | 114.7±4.1 | 96.4±3.4% |
| | 2h | 847.7±36.5 | 88.7±4.2 | 74.0±2.7% |
| | 4h | 1126.2±46.3 | 56.9±5.3 | 46.9±3.8% |
| | 8h | 1261.8±39.9 | 41.3±4.6 | 35.4±4.2% |

## 3.7 Postoperative evaluation of OCAs

The quality evaluation at 4 weeks after operation showed that the mean value of μCT of transplanted sites in each group was: control group—628.0±30.1, 2h group—854.2±23.4, 4h group—1052.6±45.3, 8h group—1158.3±43.2 HU. The GAG contents were calculated according to the relationship between μCT and GAG content as follows: control group—113.9±3.4, 2h group—88.0±2.7, 4h group—65.3±5.2, 8h group—53.2±4.9 ug/mg. Compared to the GAG content before chondroitinase treatment, the percentages of GAG content after treatment were as follows: control group—95.8±3.7%, 2h group—73.4±2.5%, 4h group—53.9±5.4%, 8h group—45.5±4.5%. Similarly, the mean values of μCT at the transplantation sites in each group at 12 weeks after operation were: control group—621.2±35.8, 2h group—847.7±36.5, 4h group—1126.2±46.3, 8h group—1261.8±39.9 HU. The GAG content was calculated according to the relationship between μCT and GAG content, with the following results obtained: control group—114.7±4.1, 2h group—88.7±4.2, 4h group—56.9±5.3, 8h group—41.3±4.6ug/mg. The percentages were as follows: control group—96.4±3.4%, 2h group—74.0±2.7%, 4h group—46.9±3.8%, 8h group—35.4±4.2% (Table 2).

## 3.8 Evaluation of functional changes in postoperative OCAs

The estimation of GAG contents using μCT imaging, histological and biomechanical analysis of OCA at the transplantation site was performed at 4 and 12 weeks after surgery to verify the effect of OCAs possessing different GAG contents on the functional regeneration of cartilage (Fig 5). The estimated GAG contents by μCT analysis showed that there was no significant change in GAG content in the control and 2h groups at 4 and 12 weeks, while the GAG content at 12 weeks in the 4h and 8h groups was lower than that at 4 weeks (Fig 5A, *p<0.05, ***p<0.001). The results of Safranin-O staining showed that there was no significant difference in histological scores between the control group and the 2h group, while the scores at 12 weeks were lower than those at 4 weeks in the 4h and 8h groups (Fig 5B, **p<0.01, ***p<0.001). Biomechanical results showed that the compressive modulus at 12 weeks was higher than that at 4 weeks in the control and 2h groups, whereas in the 8h group, the compressive modulus at 4 weeks was higher than that at 12 weeks (Fig 5C, *p<0.05).

## 3.9 Cell viability and percentage of viable cells of OCAs

At 12 weeks after surgery, the cell viability and viable cell density at the transplantation site were evaluated by live/dead staining. Dead cells were hardly observed in the live/dead stained images of the Control and 2h groups. By contrast, more dead cells were observed in the images of the 4h and 8h groups, mainly in the cartilage superficial zone. In the 8h group in particular,

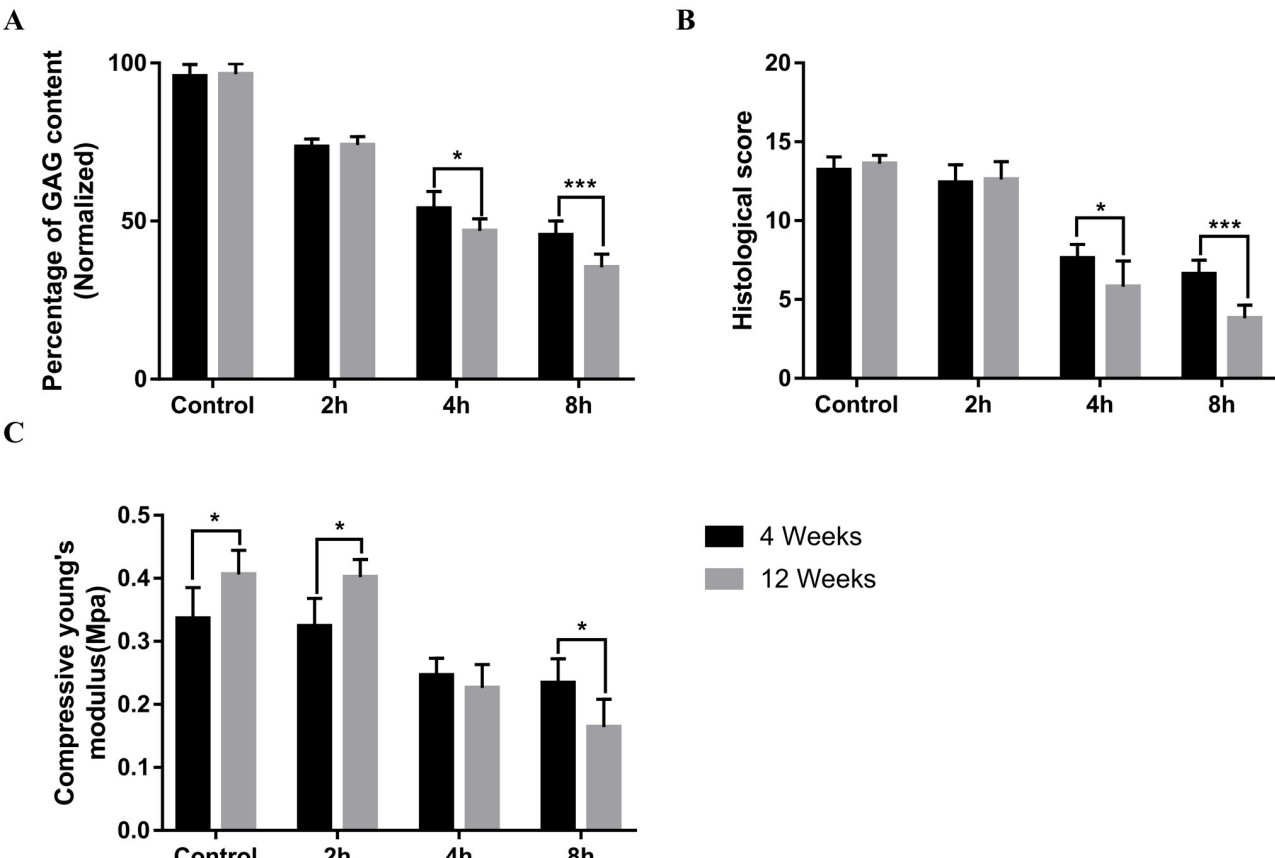

**Fig 5. Changes in OCA effect after 4 and 12 weeks of transplantation. A** Quantification of GAG content at the graft site by μCT analysis, **B** histological score and **C** compressive modulus were compared at 4 and 12 weeks postoperatively($*p<0.05$ $**p < 0.01$, $***p < 0.001$).

dead cells were observed in the middle zone (Fig 6A). Quantitative analysis was used to compare the cell viability and viable cell density. The average cell viabilities in the control group and the 2h group were 93% and 88%, with no significant difference. By contrast, the cell viabilities in the 4h and 8h groups were 55% and 33%, respectively, with a significant downward trend, particularly in the 8h group (Fig 6B, $p < 0.001$). The same trend was observed in the quantitative analysis of the viable cell density results (Fig 6C, $p < 0.001$).

## 4 Discussion

The current study evaluated the efficacy of GAG content in OCA. The Control, 2h, 4h, and 8h groups were set based on the different treatment times of graft chondroitinase. We found that the functional effects of OCA were reduced in the 4h and 8h groups compared to in the control and 2h groups, thus leading to OCA functional failure. Further, the comparison between 4 and 12 weeks after surgery showed that the GAG contents, biomechanical characteristics, and histological scores of the transplanted sites in the control and 2h groups were better maintained than those in the 4h and 8h groups. Therefore, the GAG content of the graft has the potential to be a key factor in assessing the functional success of OCA.

Hyaline cartilage is a load-bearing tissue that supports joints and evenly distributes mechanical loads. Further, because GAG is made up of polysaccharide chains and has a high

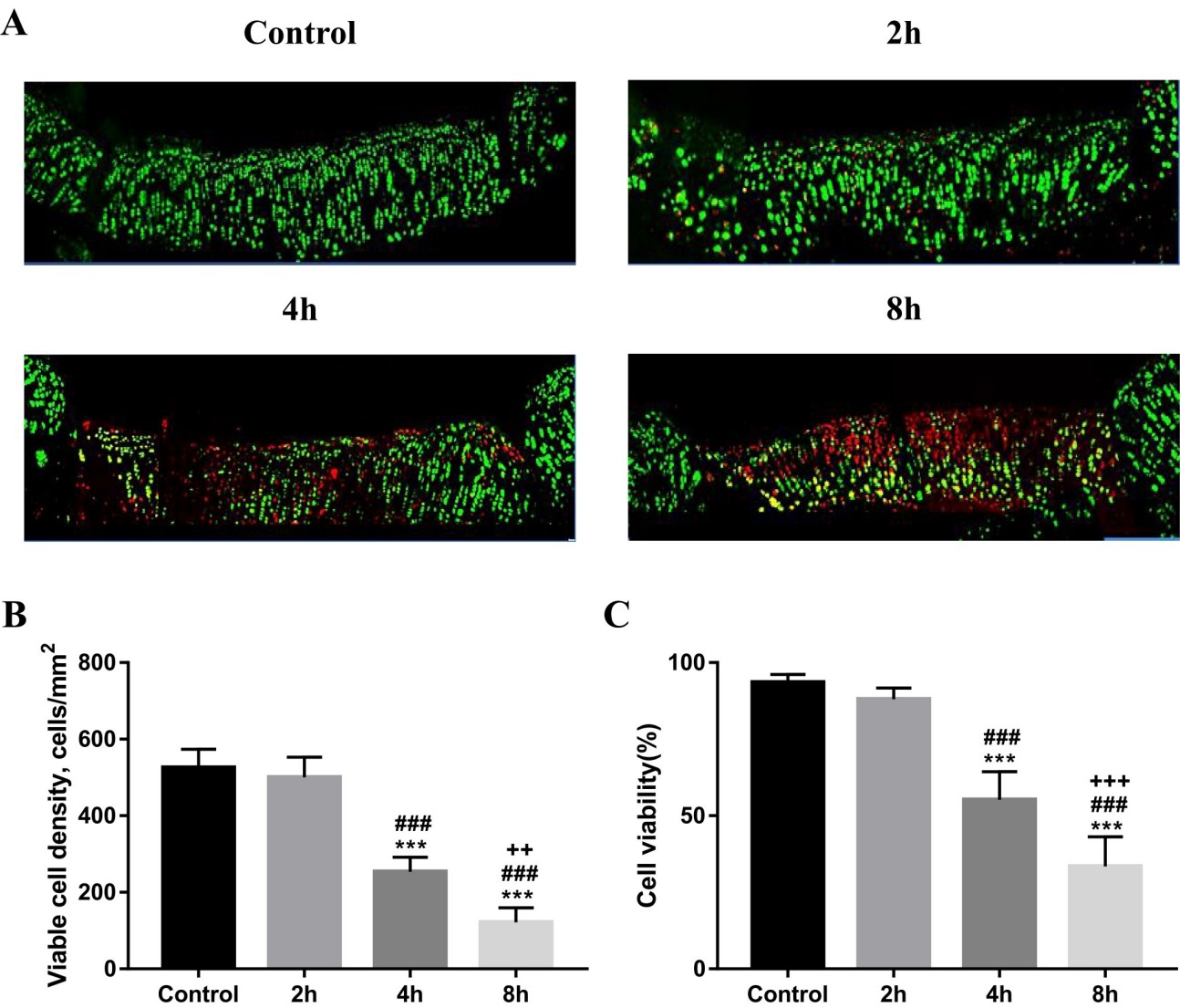

**Fig 6. Cell viability at the OCA transplantation site at 12 weeks postoperative (n = 5).** A Representative live/dead stained images of the OCA transplantation sites at 12 weeks after surgery. Dead cells are marked in red whereas live cells are marked in green. B Percentage of cell viability in transplantation site. C Quantitative analysis of the available cell density in the transplantation site ($^*p < 0.05$, $^{**}p < 0.01$, $^{***}p < 0.001$). Magnification x20.

negative charge and strong hydrophilicity, it draws osmotically active Na+, which causes a significant amount of water to be drawn into its structure. Based on the effects of osmotic pressure and chemical expansion stress caused by these GAG characteristics, the ECM can endure compressive forces [47]. When cartilage tissue is subjected to continuous pressure load, this function maintains the stability of the extracellular microenvironment, which ultimately prevents cell damage and cell apoptosis [34, 35, 37]. Levin A.S. et al. [48] proved that the GAG content in immature bovine cartilage was higher than that in mature bovine cartilage, and that the cells in the former are less vulnerable to load-induced injury. Otsuki S. et al. [40] proved that GAG loss alone did not directly lead to chondrocyte death. In response to mechanical injury, there is an immediate induction of necrotic cell death. Similarly, in our histological analysis, as the GAG content of the graft decreased (from the control group to the 8h group),

the remodeling ability and histological score of the graft site decreased at both 4 and 12 weeks postoperatively (Fig 4A and 4D). The results of the biomechanical analysis showed that the compressive moduli of the control group and the 2h group were higher than those of the 4h and 8h groups at 4 and 12 weeks after surgery, indicating that the GAG content of the graft would affect the postoperative biomechanical properties (Fig 4C). Further, in the 4h and 8h groups, as the GAG content of the graft decreased, the cell viability of the graft site decreased at 12 weeks after surgery. By contrast, in both the control group and the 2h group, the cell viability of the transplanted site remained at a high level 12 weeks after the operation (Fig 6). Therefore, we assumed that changes in graft GAG content were a key factor affecting the functional success of OCA.

It is known that GAG concentrations in cartilage decrease with age. Previous study reported that human articular cartilage was obtained postmortem from the lateral femoral condyle of 30 subjects aged 1 to 70 years, and their cartilage GAG concentrations were compared. The results showed that the GAG contents of donors older than 60 years were significantly decreased [49]. In early OA, GAG in ECM will be reduced, and this decrease in GAG content reduced the elasticity of articular cartilage to compressive loads [50]. In this study, we used chondroitinase to manufacture grafts with different GAG contents, and the differences in GAG contents and the unchanged chondrocyte viability of the grafts were demonstrated by biochemical, μCT, and cell viability analyses (Figs 2 and 3). In the in vivo study, the study period included 4 weeks of early follow-up and 12 weeks of late follow-up. Due to the high potential for spontaneous healing in the rabbit animal model, the functional effects of each group of OCA were assessed at 4 weeks postoperatively [51, 52]. Shapiro et al. demonstrated encouraging results for 3mm osteochondral defect in rabbit model, at 12 weeks postoperatively [53]. We compared the functional effects of OCA at 4 and 12 weeks postoperatively by imaging, histological, and biomechanical analyses. In the quantitative analysis of μCT, the GAG content in the 4h and 8h groups decreased significantly from 4 weeks to 12 weeks after surgery, which was the same trend as the Safranin-O staining score (Fig 5A and 5B). And in the biomechanical analysis, with the increase of follow-up time, the compressive modulus of the 8h group decreased significantly (Fig 5C). These results further demonstrated that GAG content in OCA might play an important role in osteochondral repair.

GAG in articular cartilage ECM is a strongly hydrophilic polymer composed of polysaccharide chains and is highly negatively charged [47, 54, 55]. Applying these characteristics of GAG, quantitative computed tomography (QCT) has been used to measure the concentration and spatial distribution of anionic iodine contrast agents diffused into isolated articular cartilage explants. It has also been demonstrated in various animal and human models that the μCT value is linearly correlated with the GAG content of articular cartilage [38, 41–43]. In this study, we compared the μCT image and Safranin-O-stained image of the same sample, and the results showed that the trend of the change in GAG content was the same (Fig 3A). Subsequently, we used μCT analysis and biochemical analysis to measure the GAG content of each group of OCA in both in vivo and in vitro experiments (including OCA obtained at 4 and 12 weeks after surgery) and evaluated the relationship between the two analysis methods. The results show that the μCT value of the graft is negatively correlated with the GAG content and $R^2 = 0.91$, which has a reliable value (Fig 3D). Further, after μCT analysis, the cell viability of each group of grafts was shown to remain above 95%, which further demonstrated the feasibility of this method (Fig 2C–2E). Therefore, the results indicated that the μCT non-destructive analysis method can be used to detect the content of GAG in cartilage tissues under various conditions, including normal cartilage and degenerated cartilage, which has high reliability and can be applied to the evaluation method of preoperative graft.

In the transplantation of other human tissues, the content of the extracellular matrix of the graft is an important factor, as is the reconstruction of tendons and ligaments. Several studies examining animal models of ACL allografts have shown that myofibroblasts in the grafts exert contractile forces through the extracellular matrix to directly affect the remodeling capacity of their grafts [56, 57]. The extracellular matrix in the tendon and ligament varies by function, thereby providing appropriate mechanical properties [58]. Further, using extracellular matrix grafts to reconstruct the Achilles tendon was shown to enhance Achilles tendon repair, reduce gaps, and improve biomechanical effects [59, 60]. Therefore, ECM is an important influencing factor in grafts, and our study found that GAG can affect the functional effect of OCA and may become an important biomarker.

In this study, chondroitinase ABC was used to create graft models containing different concentrations of GAG. Chondroitinase ABC is a GAG-specific hydrolase that is used to digest chondroitin sulfate in proteoglycan complexes. Previous studies have shown that it can specifically degrade the content of GAG in tissues while having little effect on other components such as collagen [61, 62]. Chondroitinase has been used in many studies to modulate GAG concentrations, including in vitro and in vivo experiments in various animal and human models. Further, the change in the treatment conditions of chondroitinase can obtain more cartilage models with different GAG contents and simultaneously little effect on chondrocytes during the treatment [40, 62, 63]. We used 1 Unit/ml concentration of chondroitinase to create three transplantation models with different GAG concentrations under different treatment time conditions. The grafts in each group were treated with chondroitinase, rinsed three times with PBS solution, and then subjected to live/dead staining experiments. The results showed that few dead cells were observed in the cartilage tissue of each group, the cell survival rate was above 95%, and the difference between the groups was not statistically significant. This indicated that the chondroitinase treatment process had less effect on tissue cell viability, which is consistent with the previous findings.

This study has the following limitations: The first limitation is the selection of experimental animals. In this study, a rabbit model was used for in vivo research. The rabbit model possessed several limitations such as self-regenerative capacity, thinner cartilage thickness, and small defect area. Secondly, we measured GAG content before and after OCA surgery by μCT noninvasive analysis method. Although the test results showed that its confidential range was high, the μCT value is an indirect quantification of GAG content which may be slightly different from its own content. Finally, we manufactured grafts with different GAG content by using chondroitinase and evaluated the effect of GAG content on the functional efficacy of OCA. However, the method of making a graft model by using chondroitinase is fundamentally different than the actual difference between donors, because differences between donors are generated under important factors such as age, BMI, and history of trauma. Therefore, it is necessary to further analyze the changes in GAG contents in human donor grafts, which could be the content of our follow-up study.

In conclusion, reduced GAG contents of grafts in the rabbit model were found to affect the functional effects of post-transplantation OCA. In this study, non-destructive analysis of μCT was conducted to evaluate the GAG content of the graft before and after surgery and to verify the effectiveness of the method. Altogether, the results indicated that this quality analysis method for the preoperative evaluation of grafts could be applied in clinical treatment.

## Supporting information

**S1 Fig. Rabbit animal model of osteochondral allograft transplantation.** (A) Graft harvested from a donor rabbit knee trochlea. (B) The size of the graft is 4 mm in diameter and 2.5 mm in

depth. (C) Fully exposed recipient rabbit knee pulley. (D) Finally, the picture after implantation of the osteochondral graft.
(TIF)

## Author Contributions

**Conceptualization:** Do Young Park, Dong Il Shin, Jin Ho Park, Byoung-Hyun Min.

**Data curation:** Yong Jun Jin, Sujin Noh, Jin Ho Park.

**Formal analysis:** Yong Jun Jin, HyeonJae Kwon, Dong Il Shin.

**Funding acquisition:** Byoung-Hyun Min.

**Investigation:** Yong Jun Jin, HyeonJae Kwon.

**Methodology:** Yong Jun Jin, Sujin Noh, HyeonJae Kwon, Dong Il Shin, Jin Ho Park.

**Project administration:** Sujin Noh, Byoung-Hyun Min.

**Resources:** HyeonJae Kwon.

**Software:** Yong Jun Jin, Sujin Noh, HyeonJae Kwon, Jin Ho Park.

**Supervision:** Do Young Park, Sujin Noh, Byoung-Hyun Min.

**Validation:** Yong Jun Jin, Do Young Park, Dong Il Shin.

**Visualization:** Yong Jun Jin.

**Writing – original draft:** Yong Jun Jin.

**Writing – review & editing:** Do Young Park, Byoung-Hyun Min.

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
