## [Decision Letter · Decision Letter 0]

16 Mar 2023

PONE-D-23-00265Effects of Glycosaminoglycan(GAG) content in donor cartilage extracellular matrix on the functional properties of osteochondral allograft(OCA) as evaluated by μCT non-destructive analysisPLOS ONE

Dear Dr. Min,

Thank you for submitting your manuscript to PLOS ONE. After careful consideration, we feel that it has merit but does not fully meet PLOS ONE’s publication criteria as it currently stands. Therefore, we invite you to submit a revised version of the manuscript that addresses the points raised during the review process. Please submit your revised manuscript by Apr 30 2023 11:59PM. If you will need more time than this to complete your revisions, please reply to this message or contact the journal office at plosone@plos.org. Please include the following items when submitting your revised manuscript:A rebuttal letter that responds to each point raised by the academic editor and reviewer(s). You should upload this letter as a separate file labeled 'Response to Reviewers'.A marked-up copy of your manuscript that highlights changes made to the original version. You should upload this as a separate file labeled 'Revised Manuscript with Track Changes'.An unmarked version of your revised paper without tracked changes. You should upload this as a separate file labeled 'Manuscript'.If applicable, we recommend that you deposit your laboratory protocols in protocols.io to enhance the reproducibility of your results. Protocols.io assigns your protocol its own identifier (DOI) so that it can be cited independently in the future. For instructions see: https://journals.plos.org/plosone/s/submission-guidelines#loc-laboratory-protocols. Additionally, PLOS ONE offers an option for publishing peer-reviewed Lab Protocol articles, which describe protocols hosted on protocols.io. Read more information on sharing protocols at https://plos.org/protocols?utm_medium=editorial-email&utm_source=authorletters&utm_campaign=protocols.

We look forward to receiving your revised manuscript.

Kind regards,

Carlos Alberto Antunes Viegas, DVM; MSc; PhD

Academic Editor

PLOS ONE

“This study was supported by the Korea Health Technology R&D Project (HI17C2191) through the Korea Health Industry Development Institute, funded by the Ministry of Health & Welfare. We have no potential conflict of interest relevant to this article.”

Reviewers' comments:

Reviewer's Responses to Questions

**Comments to the Author**

1. Is the manuscript technically sound, and do the data support the conclusions?

Reviewer #1: Yes

Reviewer #2: Yes

2. Has the statistical analysis been performed appropriately and rigorously? 

Reviewer #1: Yes

Reviewer #2: Yes

3. Have the authors made all data underlying the findings in their manuscript fully available?

Reviewer #1: Yes

Reviewer #2: Yes

4. Is the manuscript presented in an intelligible fashion and written in standard English?

Reviewer #1: Yes

Reviewer #2: Yes

5. Review Comments to the Author

Reviewer #1: Title suggestion: Effects of glycosaminoglycan content in extracellular matrix of donor cartilage on the functional properties of osteochondral allografts evaluated by micro-TC non-destructive analysis

The abstract should be a little more elaborate. The abstract should consist of 1 or 2 sentences referring to each one of the various sections of the manuscript - introduction, material and methods, results, discussion and conclusions. When presenting the results, the values with a statistically significant result of the GAG content for the different times of the groups under study must be presented.

The wrote English is very confusing. The manuscript needs a major revision of the standard of English, for example:

Line 25 - There has been a dearth (lack) of systematic research…

Line 30 and 31 - Due to the different treatment time of chondroitinase, (several groups were held) it was divided into - the control group and the 2h, 4h and 8h groups.

Line 65 – The word “cryopreservation” is repeated.

There are a few more aspects of manuscript formatting that should be corrected, such as:

- the full reference to the meaning of the abbreviations the 1st time they are mentioned,

- presentation of bibliographical references in the text.

In the anaesthetic protocol to which the rabbits were subjected, the active principles of the sedative and anaesthetic drugs used and the doses/kg, as well as the analgesics/anti-inflammatories used in the postoperative period, should be mentioned.

I think the article would be enriched if some illustrative images of the surgical protocol performed were included.

Reviewer #2: The work is very well written, it presents a clear objective that it tries to demonstrate with a very well structured material and method that gives rise to clear results, a very well written discussion and a conclusion.

The worst part of the study is the anesthesia and analgesia of the animals in the in vivo studies. It does not administer any analgesic drug in the preoperative period, does not indicate which anti-inflammatory and analgesic drugs used in the postoperative period, and uses injectable anesthesia without oxygen supplementation, and the high tendency of rabbits to hypoxemia during anesthesia without oxygen supplementation is well known.

My knowledge of English does not allow me to evaluate if the manuscript is presented in an intelligible fashion and written in standard English

6. PLOS authors have the option to publish the peer review history of their article (what does this mean?). If published, this will include your full peer review and any attached files.

Reviewer #1: No

Reviewer #2: **Yes: **Antonio González-Cantalapiedra

---

## [Author Response · Author response to Decision Letter 0]

27 Apr 2023

Reviewer 1 Comments for the Author 

1.Title suggestion: Effects of glycosaminoglycan content in extracellular matrix of donor cartilage on the functional properties of osteochondral allografts evaluated by micro-TC non-destructive analysis

Answer): Thank you for reviewing our work. We've reviewed your comments and edited the title as you suggested.

2.The abstract should be a little more elaborate. The abstract should consist of 1 or 2 sentences referring to each one of the various sections of the manuscript - introduction, material and methods, results, discussion and conclusions. When presenting the results, the values with a statistically significant result of the GAG content for the different times of the groups under study must be presented.

The wrote English is very confusing. The manuscript needs a major revision of the standard of English, for example:

Line 25 - There has been a dearth (lack) of systematic research…

Line 30 and 31 - Due to the different treatment time of chondroitinase, (several groups were held) it was divided into - the control group and the 2h, 4h and 8h groups.

Line 65 – The word “cryopreservation” is repeated.

Answer): Thank you for reviewing our work. I apologize for some unclear English expressions in the article, and we have revised the article, including the abstract, word repetition, citation of abbreviations, references, etc.

3.In the anaesthetic protocol to which the rabbits were subjected, the active principles of the sedative and anaesthetic drugs used and the doses/kg, as well as the analgesics/anti-inflammatories used in the postoperative period, should be mentioned.

Answer): For the anesthesia protocol performed on rabbits, the active ingredients and doses/kg of the sedatives and anesthetics used, as well as the analgesics and anti-inflammatory drugs used postoperatively were added. We have added instructions at lines 222-230 of the text.

4.I think the article would be enriched if some illustrative images of the surgical protocol performed were included.

Answer): Thank you for your interest in our animal model, we describe the general procedure of the surgery and a simple schematic diagram in Supplementary Figure 1.

Reviewer 2 Comments for the Author 

The worst part of the study is the anesthesia and analgesia of the animals in the in vivo studies. It does not administer any analgesic drug in the preoperative period, does not indicate which anti-inflammatory and analgesic drugs used in the postoperative period, and uses injectable anesthesia without oxygen supplementation, and the high tendency of rabbits to hypoxemia during anesthesia without oxygen supplementation is well known.

Answer): Thank you for reviewing our work. We've reviewed your comments and have done our best to answer your questions. First of all, we regret not being able to inject analgesics preoperatively. Fortunately, after our operation, we started to inject analgesics and lasted for 3 days, which should relieve the postoperative pain to a great extent. For analgesics and anti-inflammatory drugs, we have added instructions at lines 222-230 of the text.

Secondly, it is a pity that we were not able to use oxygen injection anesthesia. In many previous studies, the simple animal surgery on the knee was basically performed without supplemental oxygen. Because the operation process is simple and the operation time is short, and the amount of bleeding is less, the degree of trauma to the rabbit itself is relatively low. However, we will use supplemental oxygen anesthesia in future rabbit model animal experiments to avoid hypoxemia.

Answers to editors

We have actively improved the format of the article according to the requirements, please understand, if there is a need to improve, we will modify it as soon as possible.

---

## [Editor Report · Decision Letter 1]

2 May 2023

Effects of Glycosaminoglycan content in extracellular matrix of donor cartilage on the functional properties of osteochondral allografts evaluated by micro-CT non-destructive analysis

PONE-D-23-00265R1

Dear Dr. Byong-Hyun Min,

We’re pleased to inform you that your manuscript has been judged scientifically suitable for publication and will be formally accepted for publication once it meets all outstanding technical requirements.

Kind regards,

Carlos Alberto Antunes Viegas, DVM; MSc; PhD

Academic Editor

PLOS ONE
---

## [Editor Report · Acceptance letter]

11 May 2023

PONE-D-23-00265R1 

Effects of Glycosaminoglycan content in extracellular matrix of donor cartilage on the functional properties of osteochondral allografts evaluated by micro-CT non-destructive analysis 

Dear Dr. Min:

I'm pleased to inform you that your manuscript has been deemed suitable for publication in PLOS ONE. Congratulations! Your manuscript is now with our production department. 

Kind regards, 

on behalf of

Dr. Carlos Alberto Antunes Viegas 

Academic Editor

PLOS ONE